# 100 years of lake evolution over the Qinghai-Tibet Plateau

**Guoqing Zhang[1], Youhua Ran[2], Wei Wan[3], Wei Luo[1,4], Wenfeng Chen[1,5], Fenglin Xu[1,5], Xin Li[1]**

[1] State Key Laboratory of Tibetan Plateau Earth System, Resources and Environment, Institute of Tibetan Plateau

Research, Chinese Academy of Sciences, Beijing, China

[2] Northwest Institute of Eco-Environment and Resources, Chinese Academy of Sciences, Lanzhou, China

[3] Institute of Remote Sensing and GIS, School of Earth and Space Sciences, Peking University, Beijing, China

[4] Natural Resources and Planning Bureau, Qujing, Yunnan, China

[5] University of Chinese Academy of Sciences, Beijing, China

**Correspondences**: Guoqing Zhang (guoqing.zhang@itpcas.ac.cn) and Youhua Ran (ranyh@lzb.ac.cn)

**Abstract**: Lakes can be effective indicators of climate change, and this is especially so for the lakes over the Qinghai-Tibet Plateau (QTP), the highest plateau in the world, which undergo little direct human influence. The QTP has warmed at twice as the mean global rate, and the lakes there respond rapidly to climate and cryosphere changes. The QTP has ~1200 lakes larger than 1 km$^2$ with a total area of ~46000 km$^2$, accounting for approximately half the number and area of lakes in China. The lakes over the QTP have been selected as an

essential example for global lakes or water bodies studies. However, concerning lake data over the QTP are limited to the Landsat era and/or available at sparse intervals. Here, we extend the record to provide the comprehensive lake evolution data sets covering the past 100 years (from 1920 to 2020). Lake mapping in 1920 was derived from an early map of the Republic of China, and in 1960 from the topographic map of China. The densest lake inventories produced so far between 1970 and 2020 (covering all lakes larger than 1 km$^2$ in 14

epochs) are mapped from Landsat MSS, TM, ETM+ and OLI images. The lake evolution shows remarkable transitions between four phases: significant shrinkage in 1920−1995, rapid linear increase in 1995−2010, relative stability in 2010−2015, and further increase in 2015−2020. The spatial pattern indicates that the majority of lakes shrank in 1920−1995, and expanded in 1995−2020, with a dominant enlargement for central-north lakes in contrast to contraction for southern lakes in 1976−2020. The time series of precipitation between

1920 and 2017 indirectly supports the evolution trends of lakes identified in this study. The lake data set is freely available at http://doi.org/10.5281/zenodo.4678104 (Zhang et al., 2021).

**Keywords**: lake, map of Republic of China, topographic map, Landsat, remote sensing, Qinghai-Tibet Plateau

## 1 Introduction

The Qinghai-Tibet Plateau (QTP) is the highest plateau in the world, and has a large number of lakes widely distributed across it. The QTP is sensitive to climate change: between 1970 and 2018 it warmed faster than other continental areas, with a warming rate of ~0.36 °C/decade compared to the global mean of ~0.19 °C/decade (Gistemp-Team, 2019). The warmer climate in the region during the last half century (Kuang and Jiao, 2016) has induced dramatic changes in both the hydrosphere and the cryosphere (Chen et al., 2015).

Cryospheric melting reflects in accelerated glacier retreat and ice loss (Yao et al., 2012; Shean et al., 2020; Hugonnet et al., 2021), a lower snowline (Shu et al., 2021), permafrost thawing and degradation (Ran et al., 2018), and snow melt (Pulliainen et al., 2020), which feeds crucial water to alpine lakes. Human effects on these alpine lakes' evolution can be considered to be negligible, due to their remoteness and the harsh weather conditions. These lakes respond rapidly to climate and cryosphere changes, as most of them are located in

closed watersheds (basins), and they have been selected as typical examples in studies of global water bodies (Pekel et al., 2016) and lakes (Woolway et al., 2020).

Changes in lake area can have important influences on terrestrial ecosystems and climate change due to the change between water and land. Knowledge of lake changes over the QTP has been greatly improved by the application of remote sensing techniques which allow changes in lake area, level and volume to be derived

from satellite data. Lake mapping and the determination of changes in lake number and area (Zhang et al., 2020b; Sun et al., 2018) are the most extensively researched applications of these techniques on the QTP. Lake mapping from satellite data can acquire data for lakes with an extensive range of areas (greater than 4 pixels in size) and for long monitoring periods (Ma et al., 2010; Wan et al., 2014).

Several studies have evaluated the potential and accuracy of maps of the Republic of China (from 1912 to

1949). For example, Han et al. (2016) compared the changes of rocky desertification in Guangxi province from a 1930s topographic map (with a scale of 1:100 000) and Landsat TM in ~2000. Kong (2011) evaluated the accuracy (offset) of the early Republic of China map of Henan Province (1:100 000). Su et al. (2018) provided a lake and wetland data set for Xinjiang of late Qing and Republican China (~1909 and 1935). Yu et al. (2020) described spatial-temporal evolution of Dongting lake in Hunan Province, southeastern China since the late

Qing Dynasty using a topographical map from the Republican period. All of these studies confirm the value of

the early maps in geoscience science, as a historical and rare archive. However, no studies have yet reported

lake mapping for the remote QTP in the Republic of China (the early 20th centuries).

Changes in present lake number and area (mainly satellite era, 1960s−) have been considered in several

studies: 1) Lake inventories from the 1960s topographic mapping (Wan et al., 2014), Landsat images in 2010

(Zhang et al., 2019), and Chinese Gaofen-1 satellite data in 2014/2015 (Wan et al., 2016; Zhang et al., 2017a);

2) changes in lake number and area between two or multi-phases (Zhang et al., 2019); 3) changes in lake area

for some selected large lakes with continuous satellite observations (Wu et al., 2017; Lei et al., 2013; Zhang et

al., 2020b); 4) changes in area for dominant lake distribution regions, such as the Inner Plateau (Qiangtang

Basin), between periods such as 1970s−2011 (Song et al., 2013) and 2009−2014 (Yang et al., 2017). Global

surface water bodies have been mapped by the Google Earth Engine (GEE) platform (Pekel et al., 2016).

However, the lakes in that study are defined as permanent water surfaces (i.e. those which persist throughout

the year) (Pekel et al., 2016). A detailed classification of water bodies (including lakes, reservoirs and rivers)

and precise mapping of their number and area are important for discovering surface water characteristics and

their changes. We emphasize that lakes should be differentiated carefully from other water body types, such as

rivers and reservoirs, when examining their changes (Zhang et al., 2020a), and they should be mapped in a

relatively stable season (Sheng et al., 2016). Therefore, lakes cannot be extracted directly from this global

water body data set.

Here, we provide the most comprehensive lake mapping yet produced over the QTP covering the past 100

years (from 1920 to 2020). The new features of this data set are: 1) its temporal length - it provides the longest

period of lake observations from maps; 2) it provides a state-of-the-art lake inventory for the Landsat era (from

the 1970s to 2020); 3) it provides the densest lake observations for lakes with areas larger than 1 km$^2$.

## 2  Study area

The QTP, with an area of ~$200 \times 10^4$ km$^2$, consists of Tibet Autonomous Region and Qinghai province, and

has the border of China as its southern boundary (Figure 1). The boundary of QTP is different from the usually

named as the Tibetan Plateau (TP, the area with an altitude above 2500 m a.s.l.) (Figure S1).  For the creation

of the lake inventory from the historical maps, we use the QTP, rather than the TP, as the map of the early

Republic of China and the topographic map of China do not include areas outside China. For the post-1970s Landsat era, we provide lake mapping for both the QTP and the TP as the Landsat images provide global coverage.

There are 70 China Meteorological Administration (CMA) weather stations in the QTP. The data from these stations reveal that the annual warming rate between 1980 and 2018 was $0.05 \pm 0.01$ °C/yr ($P<0.0001$) (Figure S2). The annual precipitation also shows an increasing trend, especially after 1998 (an increase of ~7% in 1999−2018 relative to 1980−1998). The QTP includes 87% of the lakes of the TP and 92% of their area. The number of lakes larger than 1 km$^2$ is ~1200, and they have a total area of ~46000 km$^2$ (Figure 1). The glacier area in this region, obtained from the second Chinese glacier inventory (Guo et al., 2015), is ~$3\times10^4$ km$^2$. The area of clear glacier shrank by ~0.08%/yr in Inner Tibet between 1990 and 2018 (Huang et al., 2021).

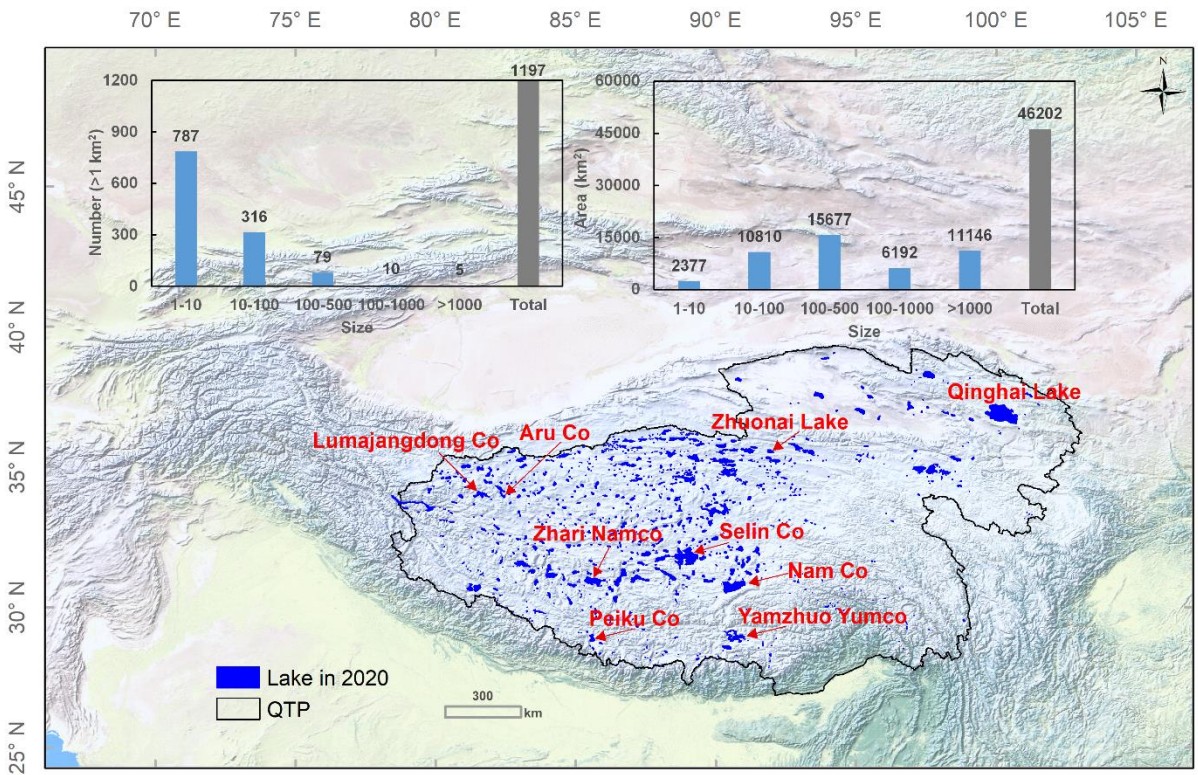

**Figure 1.** Distribution of lakes over the QTP. Insets show, for lakes larger than 1 km$^2$, the number and area of lakes in different size ranges and the total number and area (status: 2020). The names refer to the lakes presented as examples in Figure 7. The background of this figure is from Natural Earth at https://www.naturalearthdata.com.

## 3 Data and methods

Three different data sources were used: the early map of the Republic of China in ~1920 (hereafter 1920

used), a digitalized topographic map of China from the ~1960s (hereafter 1960), and Landsat

MSS/TM/ETM+/OLI images from 1970−2020. The lake mapping process for the period 1920 to 2020, shown

in Figure 2, consists of three main steps: data collection, preprocessing, and water body classification.

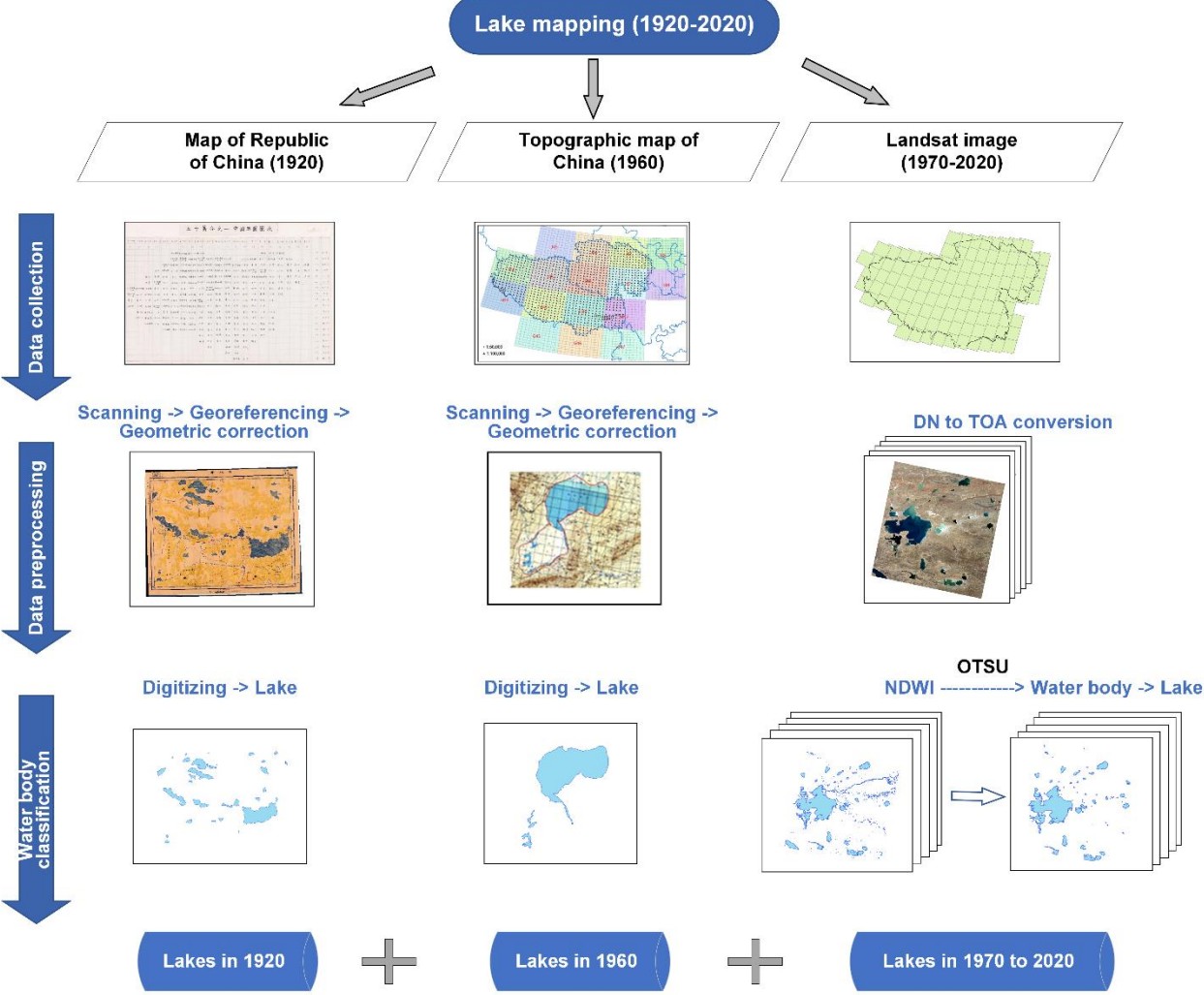

**Figure 2.** Flowchart showing the process for lake mapping of the QTP from 1920 to 2020. Three key steps: data

collection, data preprocessing, and water body classification for maps of the early Republic of China in 1920,

topographic map of China in 1960, and Landsat images in 1970−2020 are shown. The distribution of topographic

maps over the QTP is from Wan et al. (2014).

### 3.1 Lake mapping in 1920

Maps of the Republic of China used in this study are from "1/500 000 map of China", which is mainly surveyed and drawn by Beijing Army Survey Bureau, Staff Headquarters Cartographic Bureau, and Guangdong Army Survey Bureau in 1916−1918 (http://www.ccartoa.org.tw/news/2018/180901.html) (Table 1). The Army Survey Bureau of the Republic of China made the "Ten-Year Rapid Survey Plan", which required all provinces to complete topographic maps of different scales as soon as possible. The Maps of the

Republic of China were finished at beginning to apply surveying and mapping techniques, which was the first national basic surveying and mapping program after the establishment of the General Administration of Army Surveying of the General Staff Headquarters of the Republic of China in 1914. The maps were compiled with $1^\circ$ in latitude by $1^\circ$ in longitude square grid, a regional independent (hypothetical) coordinate system in each province such as polyhedral projection (modified multiconic projection), and multicolor printing. The actual

measurement of astronomy, triangulation, and barometric altimetry were utilized to provide a plane and elevation measurement control basis for topographic survey. This is the most accurate topographic map with the most complete coverage available from the early 20th century. About 50 maps with a scale of 1:500 000, covering the entire QTP, were obtained from Academia Sinica (https://www.sinica.edu.tw/ch). Each map was scanned and georeferenced. A geometric correction was also applied to remove distortion introduced by the

scanning process. Manual digitizing processes were used to vectorize water bodies from the raster maps. 604 lake water body units greater than 1 km$^2$ each over the QTP were identified, and annotated with names (old names from the early map and current Chinese names).

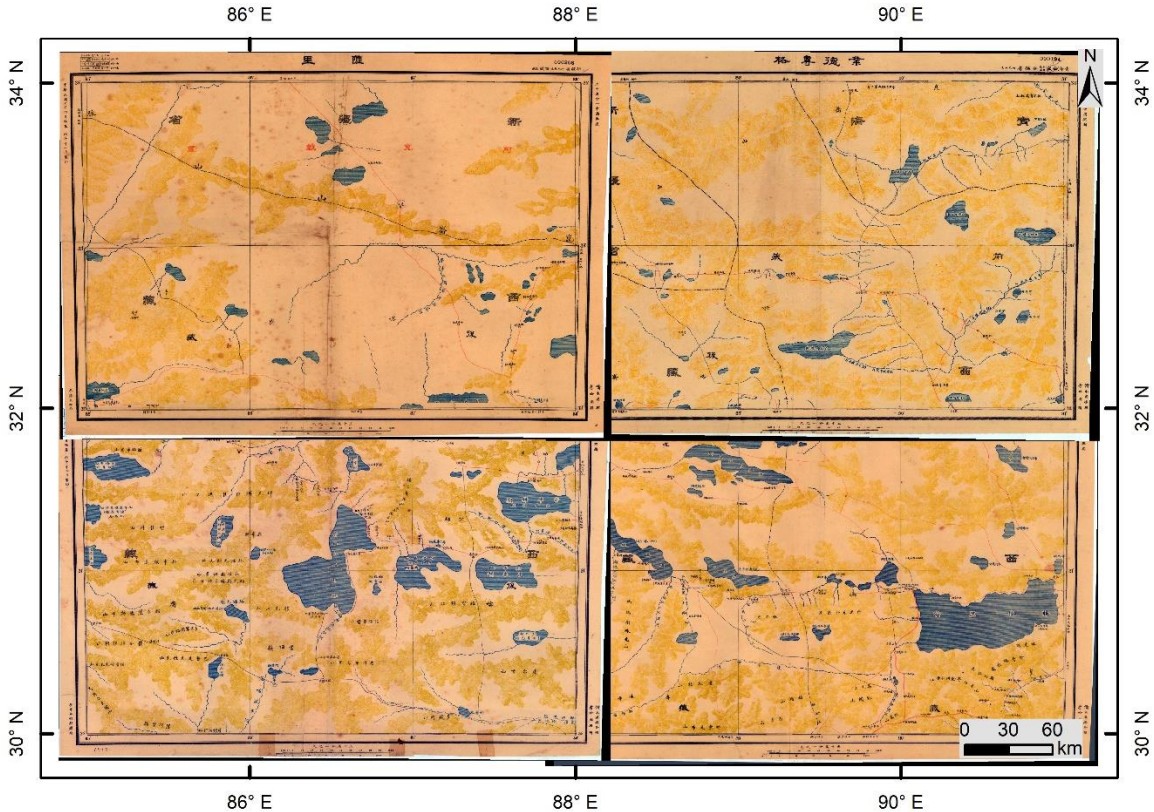

**Figure 3.** An example showing lake distribution in 1920 on maps of the Republic of China with georeferencing.


### 3.2 Lake mapping in 1960

The first Chinese lake inventory by field investigation and topographic maps created from aerial photographs surveyed mainly in the 1960s (with a scale of 1:250 000) were obtained (Table 1) (Wang and Dou, 1998; Wan et al., 2016). All maps were georeferenced to an Albers Equivalent Conical Projection, with an

Root Mean Square (RMS) error of <1.5 pixel (Ma et al., 2010). Visual interpretation and manual digitization were chosen to vectorize lake boundaries. Salt crusts, as found around the salt lakes in the northeastern QTP, were not considered to be a component of the lakes. All lakes were coded along with other attributes (names and area).


**Table 1.** Maps (images) and methods used for lake mapping from 1920 to 2020.

| Map | Sense | Period | Resolution | Number | Method for lake mapping |
|---|---|---|---|---|---|
| Early Republic of China | - | 1920s | 1:500,000 or ~130 m | ~50 | Manual digitization |
| Topographic map of China | - | 1960s | 1:250,000 or ~75 m | - | Manual digitization |
| Landsat | MSS | 1970s | ~60 m | 93 | Manual digitization |
| | TM, ETM+, OLI | 1990−2020 | 30 m | 1384 | NDWI + Otsu |

### 3.3 Lake mapping in 1972−2020

Remotely sensed images at a high spatial-temporal resolution from the Landsat series of satellites provide the longest data set (~50 years) of global lake observations (Wulder et al., 2019). The initial Landsat-1 Multispectral Scanner System (MSS) launched in 1972 provides land surface observations with a resampled resolution of ~60 m. The Thematic Mapper (TM) on Landsat-4 (launched in 1982) and -5 (launched in 1984) has a spatial resolution of 30 m for the six reflective bands, which offers a significant advance in spatial,

spectral, and geometric performance (Chander et al., 2009). The Enhanced Thematic Mapper Plus (ETM+) onboard Landsat-7 launched in 1999 has the same spatial resolution as the TM sensor. Unfortunately, the scan-line corrector (SLC) of the Landsat 7 ETM+ sensor failed in 2003, resulting in ~22% pixels missing from each scene (Chen et al., 2011). The Operational Land Imager (OLI) on Landsat-8 which launched in 2013 included a significant improvement of image radiometric quality providing better surface water mapping (Masek et al.,

2020; Wulder et al., 2019; Roy et al., 2014). A policy of open and free access to archived and new Landsat images operated by the United States Geological Survey (USGS) since 2008 has led to a substantial increase in Landsat applications (Zhu et al., 2019). However, the generation of temporally-continuous lake mapping over the QTP can still be a challenge due to the contamination of these optical images by high amounts of cloud cover and snow (Yu et al., 2016).

Landsat Level 1 terrain-corrected (L1T) products from the MSS/TM/ETM+/OLI sensors between 1972 and 2020 were downloaded from https://glovis.usgs.gov. This product is defined as the Universal Transverse Mercator (UTM) projection with World Geodetic System 84 (WGS84) datum. Radiometric calibration and a systematic geometric correction have been applied to Landsat L1T products before their release.

It is not possible to map lakes at an annual interval over the QTP due to the high levels of cloud

contamination of the Landsat images and because of seasonal fluctuations. Here, our purpose is to cover all

lakes (with area larger than 1 km$^2$) in each epoch. Lakes of area less than 1 km$^2$ are also selected for water body

classification so that no lakes are missed. The images in seasons with the largest and relatively stable lake area

are preferentially chosen (Sheng et al., 2016). An evaluation of the seasonal lake area cycle, and the presence of

relatively low levels of cloud coverage, identifies October as the optimal season over the QTP (Zhang et al.,

2020c). If there are no images available for October, we widen the search to September and November, and so

on. This process leads to the selection of some images from the winter months, but the number is small and

none contain lake ice. This time-window extension is required particularly in the early Landsat period (before

1990) for which there are few available images and data quality is low. For lake mapping in a specific target

year, the data in the optimal season for this year are first selected, and then, if not available, have a slight

extension in a preceding year. If there is still not enough data, images from the subsequent year are also used.

With this procedure, a total of 1477 images between 1972 and 2020 were used to provide 14 epochs of lake

mapping (for lakes >1 km$^2$) (Table 1). We use 1970 for the 1970s, 1990 for the 1990s, 1995 for the period

1994−1996, 2000 for the 2000s, 2005 for the period 2003−2006, and 2010 for the 2010s to better present the

spatial-temporal changes in the lakes and to match with the time series of climate data.

A summary of the images used for each epoch is presented in Figure 4, which shows the year and month

in which the images are acquired. In the 1970s, 93 images from Landsat MSS between 1972 and 1977 were

selected, with most of the images from 1976. In the 1980s, there are few available Landsat data over the QTP

and no alternative data sources, so there is a gap in the data at this time. For lake mapping in the 1990s, data

from 1990 were the first choice. Data from the preceding period, 1987−1989, and the subsequent year, 1991,

were also selected to reduce seasonal anomalies. This choice resulted in 99 images spanning five years. For

lake mapping in 1995, data from 1995 were the first choice for selection, but to meet the requirements the data

selection period had to be extended to 1994 and 1996, finally allowing 97 images to be chosen. For the 2000s,

91 images, predominantly from 2000 but with a small extension to the preceding years 1998 and 1999, were

selected.  For the 2005 lake mapping, 93 images were collected including 64 (69%) in 2005, 10 in 2003, 18 in

2004, and an additional scene in 2006. For the 2010 lake mapping, most of the data (75%) was from 2010, with

an additional 22 images (25%) from 2008 and 2009. After 2013, Landsat 8 provides more available data enabling annual lake mapping to be achieved between 2013 and 2020. For this period about 120 Landsat images in each year are required to cover the whole QTP. The small variations in this number from year to year are due to cloud coverage over some lake surfaces which leads to more data having to be employed.

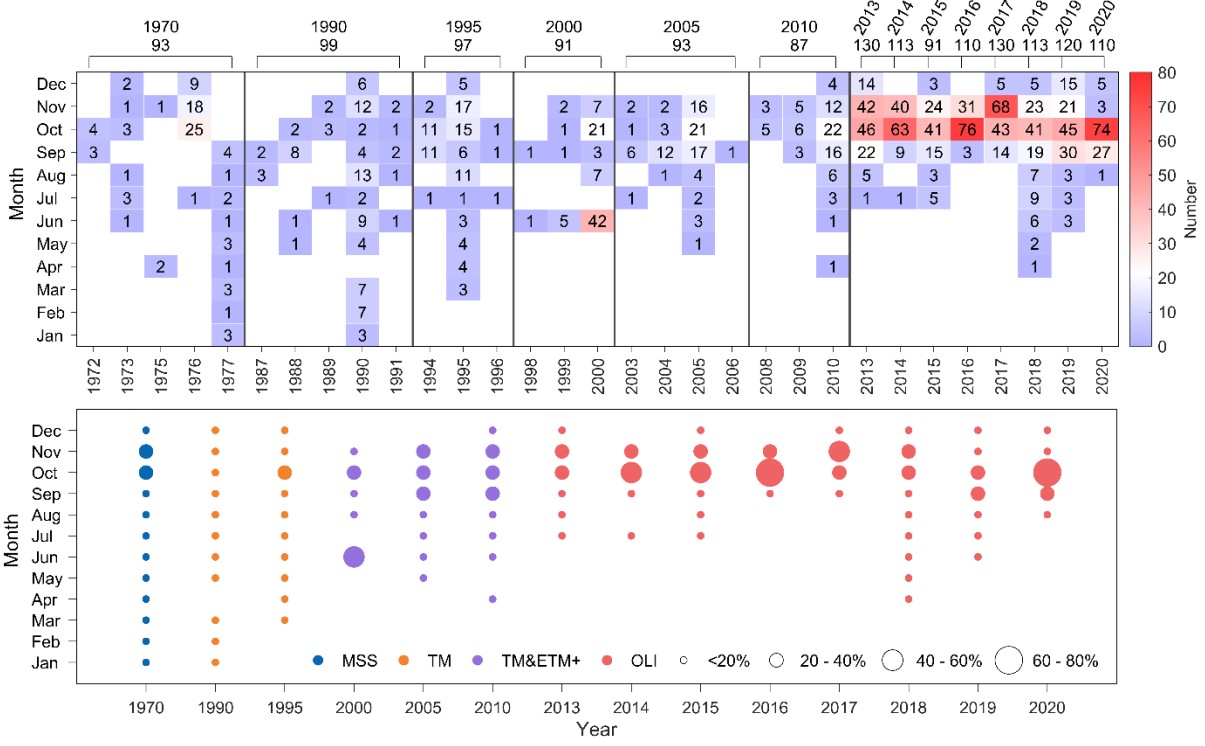

**Figure 4.** Landsat images used for lake mapping from 1970 to 2020. a) Data span in different years for each epoch, and number of months of data acquired. b) Images used from different sensors, and percentage acquired in each month.

Visual interpretation was used for lake boundary delineation in 1970 due to the low quality of the MSS data. From 1990 onwards, a semi-automated water-body classification method was used to distinguish water from non-water features. First, the Digital Numbers (DNs) of raw Landsat TM/ETM+/OLI images are converted to Top-Of-Atmosphere (TOA) reflectance (Chander et al., 2009), which is often used to generate the normalized difference water index (NDWI), a binary water / non-water classification (Mcfeeters, 2013; Li and

Sheng, 2012). Second, an automated water extraction process is executed (Zhang et al., 2017a). Initially, a

small threshold, such as "0", was used to separate water from non-water features. An outward buffer zone is then created for each water body, and an optimal threshold was determined by the Otsu method (Otsu, 1979). The water / non-water classification is performed again using the optimized threshold to generate the water body products. Finally, the lake boundaries are extracted, visually compared with the original Landsat images,

and manually edited if any mismatches are found.

### 3.4 Validation of lake mapping

A combination of direct and indirect validation was used to assess the accuracy of lake boundaries determined from the three different data sources. The lakes determined from the 1920 map of the Republic of

China and the 1960 topographic map of China cannot be validated directly as there are no records of field measurements of lake boundaries for the relevant periods. However, for these, we can indirectly compare with time series of precipitation changes (see Section 5.2) as the plateau lakes are dominantly driven by variations in precipitation (>70% contribution to lake water balance) (Biskop et al., 2016; Zhang et al., 2020b). Moreover, the veracity of the 1960 topographic maps has been demonstrated by previous studies (Wan et al., 2016; Ma et

al., 2010; Wang and Dou, 1998).

Landsat images have been widely used for water body classification and lake water-extent mapping globally (Pekel et al., 2016; Pickens et al., 2020; Verpoorter et al., 2014; Yamazaki et al., 2015). Here, we further confirm the accuracy of the water classification algorithm both directly and indirectly. High-precision GPS measurements of the boundaries of Aru Co in the northwestern plateau were made on July 28, 2017

(Figure 5). The Landsat-8 image for July 29, 2017 was acquired for the purpose of comparison with these measurements, and Figure 5b shows that the lake boundary delineated by the method of this study matches well with the data from the GPS survey. In addition, we compared the area extracted for Qinghai Lake with the lake level recorded by a gauging station between 1976 and 2019. A well-established relationship ($r^2$=0.95) between lake area and water level was exhibited (Figure 5d), indirectly suggesting that our algorithm for lake boundary

determination provides a true reflection of lake state.

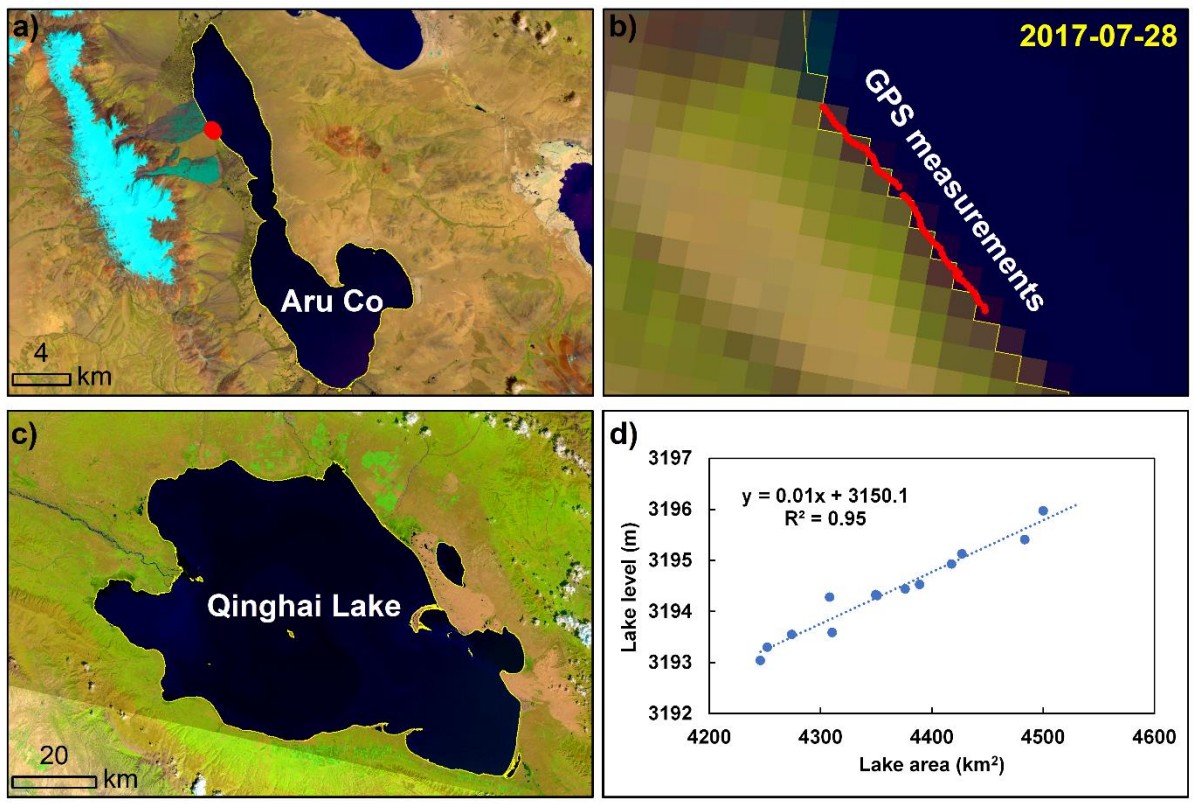

**Figure 5.** Validation of delineated lake boundaries from GPS measurements and comparison with lake level, with background of Landsat images. a) Location of GPS measurements points for Aru Co. b) Distribution of GPS measurements points along the water boundary of Aru Co. c) Boundary of Qinghai Lake from Landsat image in 2020. d) Comparison between lake area and water level from gauging station for Qinghai Lake between 1976 and 2019. Locations of Aru Co and Qinghai Lake on the QTP are shown in Figure 1.

### 3.5 Meteorological data

Annual temperature and precipitation data (~1 km resolution) between 1920 and 2017 over the QTP were derived from Peng et al. (2019). This data set is spatially downscaled from the Climatic Research Unit (CRU) data, and shows good agreement with CMA weather stations across China between 1951 and 2016. In addition, changes in annual precipitation between 1980 and 2018, determined from 70 CMA weather stations, were also utilized to indirectly compare with lake area variations (Figure S2).

### 4   Results

**4.1 Past and present lake states**

Lakes larger than 1 km$^2$ in 1920, 1960, and 1972−2020 were extracted. For 1920, a total of 604 lakes over the QTP were identified, with areas ranging from 1.06 to 4303.40 km$^2$, and a total area of 40779.33 km$^2$ (Figure 6). The number of lakes larger than 1 km$^2$ (from 1.0 to 4284.13 km$^2$) increased to 1107 in 1960, but their total area decreased to 38314.51 km$^2$, a 6% drop relative to the area in 1920. By 2020, the number of lakes larger than 1 km$^2$ increased to 1197 with a total area of 46201.62 km$^2$ (21% higher than the area in 1960). Out of the ~1200 lakes in 2020, 787 lakes (2377 km$^2$ in total area, 5%) are within the size range of 1−10 km$^2$ (inset of Figure 1). Out of 410 lakes with areas greater than 10 km$^2$, 5 are larger than 1000 km$^2$ (11146 km$^2$ in total area, 24%). In total, the number of lakes increased by ~98% and the area by ~13% between 1920 and 2020. The great increase in number may be attributed to the appearance of new lakes such as those detected by multi-period satellite images (Ma et al., 2010; Zhang et al., 2020b).

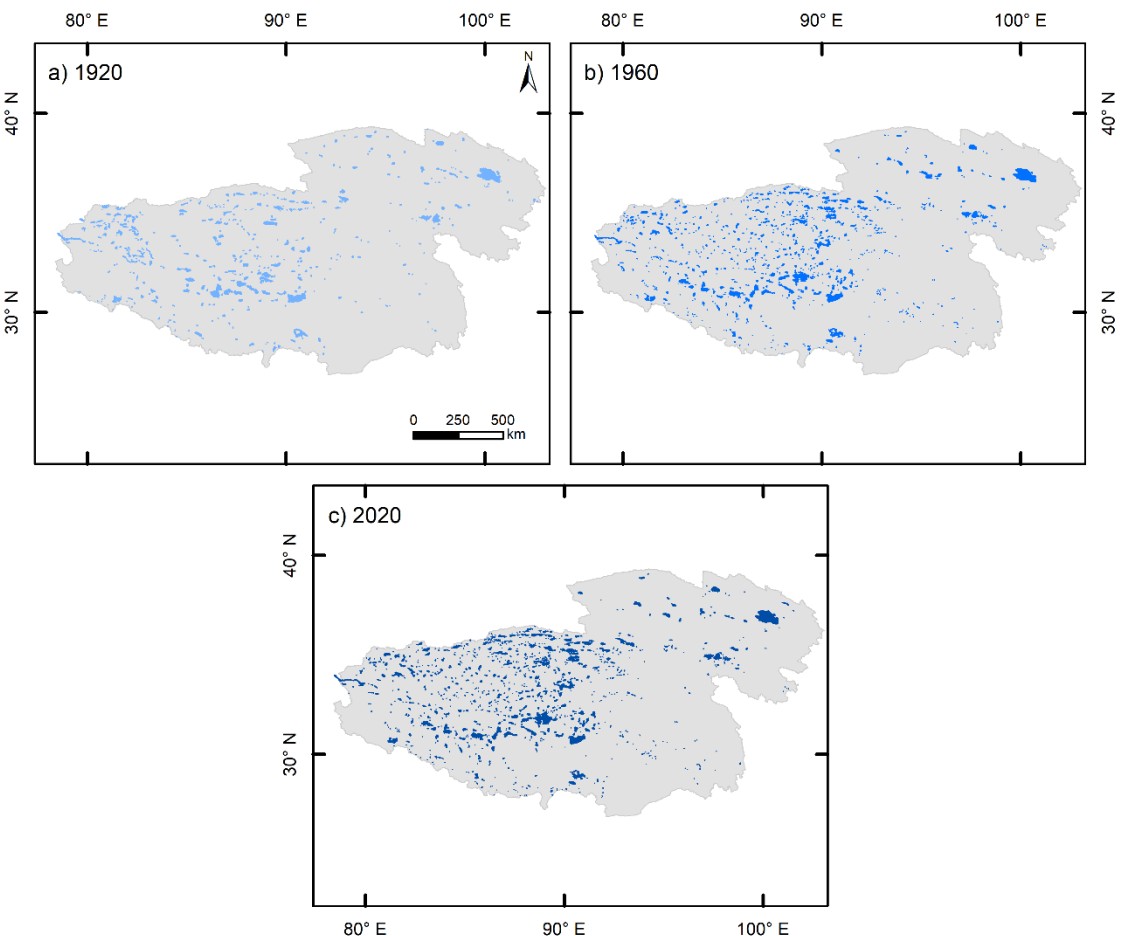

**Figure 6.** Mapped lakes in 1920 from maps of the Republic of China, 1960 from Topographic map of China, and 2020 from Landsat 8 images.


Individual lakes exhibit different patterns of evolution. Eight large lakes distributed in different climatic regions (Figure 1) have been selected to display their distinguishing characteristics (Figure 7a). Qinghai Lake, the largest lake on the QTP, had a relatively stable area before 2010 (except for a fluctuation in 1976), but a continuous rapid expansion in the most recent decade. Selin Co, the largest lake in Tibet, exhibits a robust

expansion during the past 100 years, especially after 2000. In the Landsat era, the area expanded by ~40% (Figure 7b). This study extended the record to early 1920 and found a triple increase in area has occurred. Nam Co, the second largest lake in Tibet, shrank in area before 1960, and then remained in a relatively stable state until 2020. Lumajangdong Co, in the northwestern plateau, expanded monotonically between 1920 and 2020 with an overall ~55% increase in area. The graph for Zhari Namco in the central plateau reveals a rapid

expansion before 1960, followed by a stable state, or slight increase (~6%), between 1960 and 2020. Zhuonai Lake in the northeastern plateau expanded until 1960, and then remained in a stable state between 1960 and 2010. However, an outburst flood event occurred on September 15, 2011 (Liu et al., 2016), resulting in a rapid lake water retreat as indicated by the area shrinkage between 2013 and 2020 (Figure 7b). Two large lakes in the southern plateau, Paiku Co and Yamzhuo Yumco, have undergone a slight lake level decline according to

altimetry data or observations of area shrinkage from Landsat images (Phan et al., 2012; Zhang et al., 2020b; Li et al., 2019). Here, it is shown that the shrinkage in area for these two lakes can be traced back to 1920, with decreases in area of ~22% for Paiku Co and ~49% for Yamzhuo Yumco, found between 1920 and 2020.

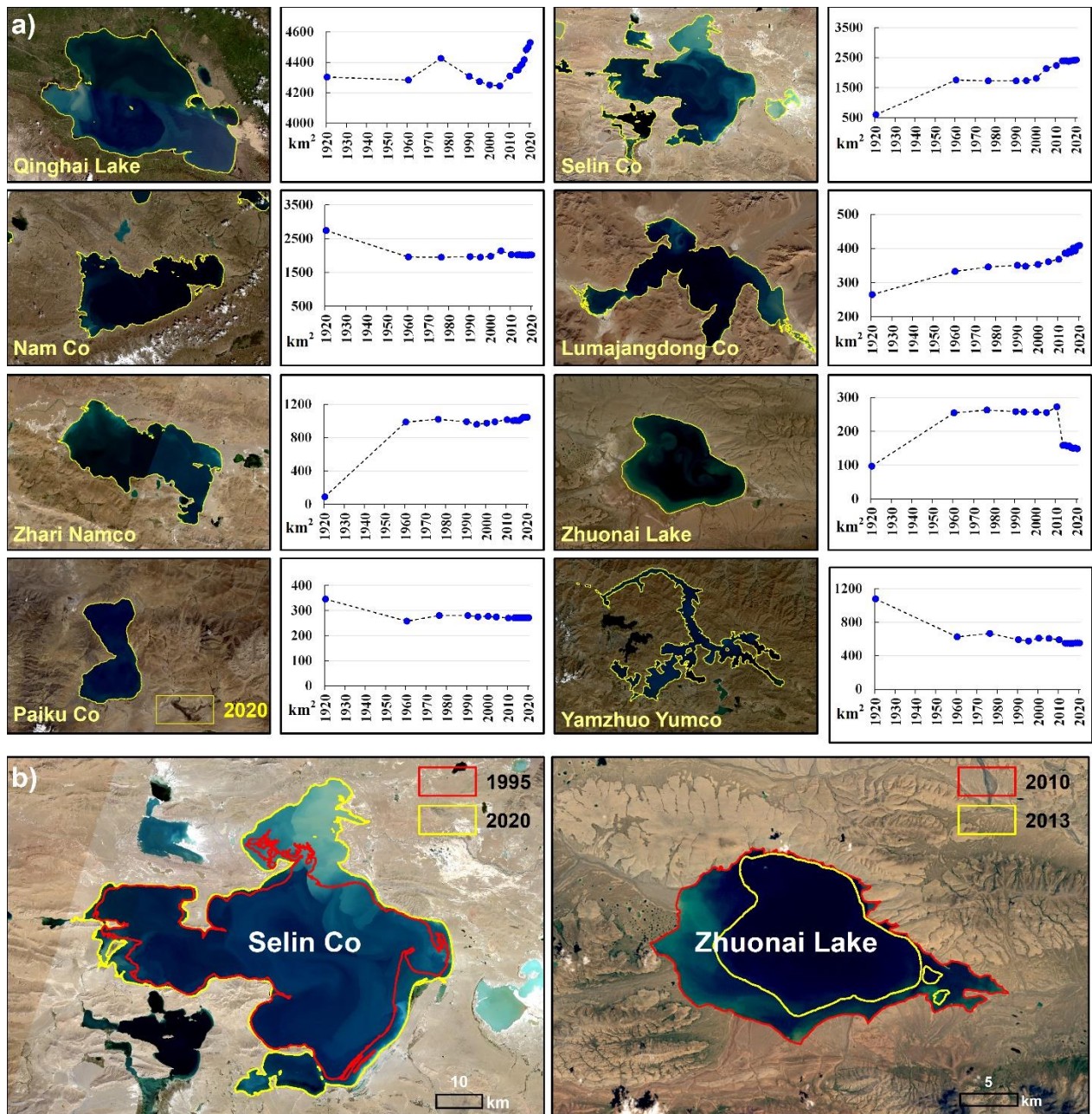

**Figure 7.** Examples showing the evolution of eight selected lakes from 1920 to 2020, with background of Landsat

images (Bands 3, 2, 1 composite). a) Time series of eight selected lake area from 1920 to 2020. b) Images for Selin

Co and Zhuonai Lake as examples of rapid lake expansion and rapid shrinkage, respectively. The locations of these

lakes are shown in Figure 1.

**4.2 Time series of lake changes between 1920 and 2020**

Here, the long-term evolution of lakes with areas larger than 1 km$^2$ between 1920 and 2020 over the QTP are described. Over the period 1920−1995, total lake area showed a remarkable 13% reduction, from 40779.33 in 1920 to 35308.31 km$^2$ in 1995 (Figure 8a). Between 1995 and 2010, lake area increased rapidly in a linear fashion, almost recovering to its initial 1920 value by 2005 and reaching a value of 43194.82 km$^2$ in 2010 (+22%). However, a more stable period with a slight shrinkage occurred between 2010 and 2015. The most

recent five years (2015−2020) were another period of rapid expansion in lake area.

Since the number and area of lakes of small size are variable, we also examined the trends in number and area of large lakes of more than 100 km$^2$ in area. The patterns of evolution of number and area for these large lakes (Figure 8b) are consistent with the total area of all lakes greater than 1 km$^2$, supporting the several different lake evolution phases during the past 100 years identified above.

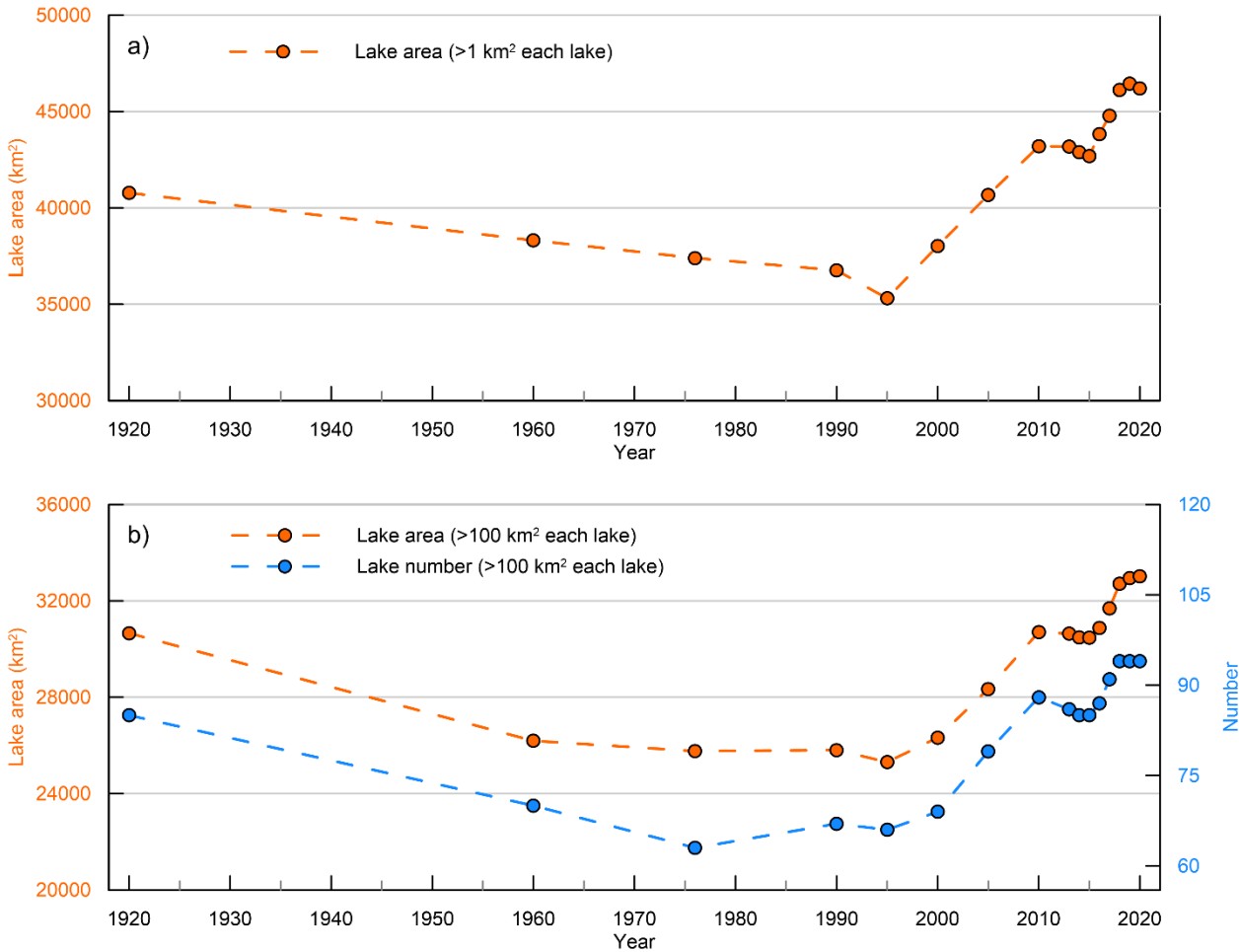


**Figure 8.** Changes in number and area of lakes from 1920 to 2020. a) Total area of all lakes greater than 1 km². b) Total area and number of lakes greater than 100 km² to exclude the effects of variability in the number and area of small lakes (1−100 km²).

### 4.3 Spatial patterns of lake changes between 1920 and 2020

The spatial patterns of lake changes for three different periods are presented in Figure 9. Between 1920 and 1995, most of the lakes throughout the plateau shrank in area. However, almost all of the lakes increased in area between 1995 and 2020. Over the entire study period, 1920−2020, the spatial pattern is heterogeneous, with the majority of shrinking lakes in the central-south plateau, but most of the enlarging lakes in the northern plateau.

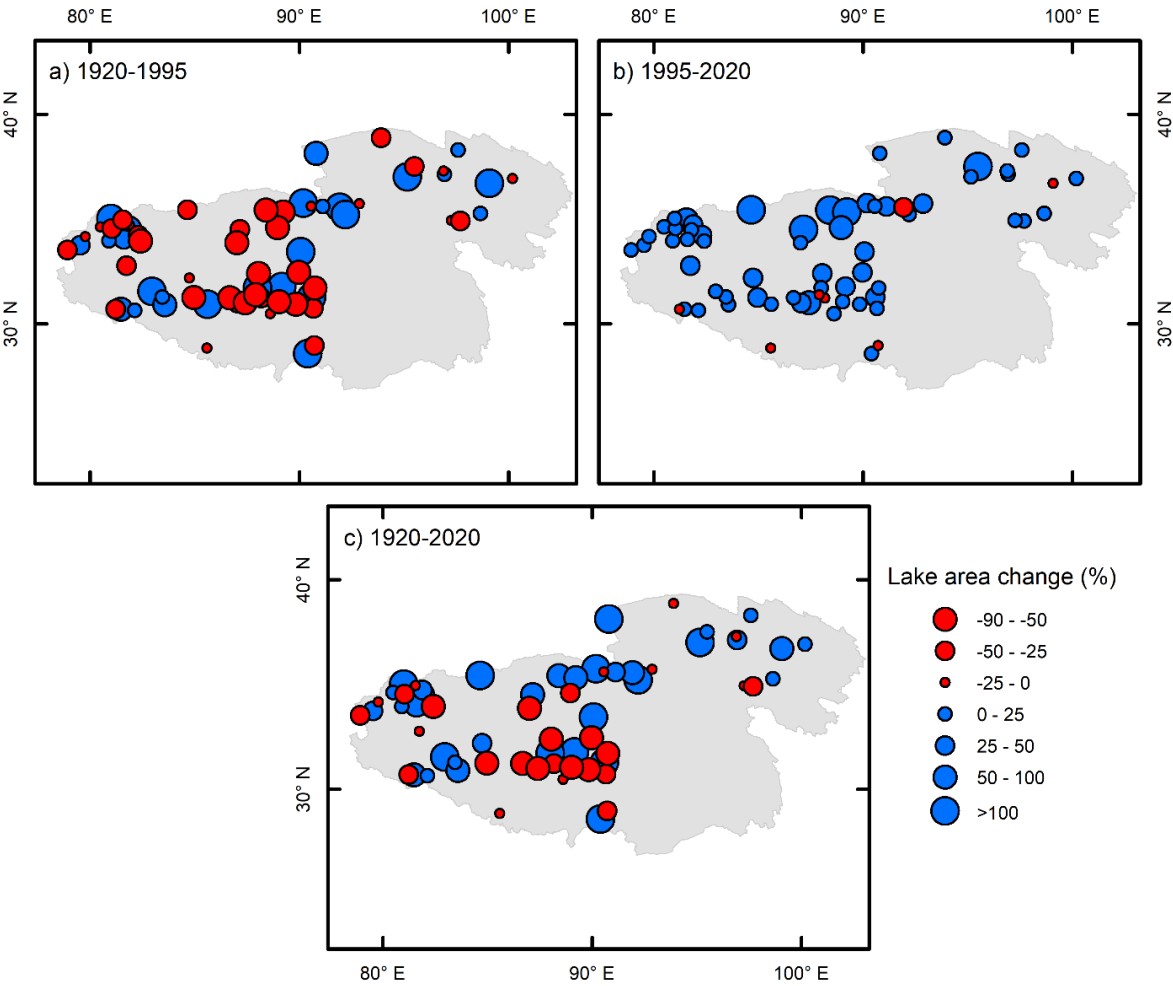

**Figure 9.** Spatial patterns of lake area changes during different evolution stages. a) 1920−1995. b) 1995−2020. c) 1920−2020.

The spatial patterns of lake area changes during the Landsat era are further examined (Figure 10) as Landsat images are used popularly, and the past five decades cover the period with the most significant lake changes. Between 1970 and 1995, the majority of lakes decreased in area. However, this pattern was reversed during 1995−2010 when the vast majority of lakes grew in area. In the most recent stage, 2010−2020, most of lakes were still enlarging, although there were some contracting lakes in the central plateau. Over the entire

period of 1970−2020, the predominant lake behavior was expansion, but some, mainly southern, lakes shrank. One such exception is Zhuonai Lake (Figure 10c−d) which underwent a noticeable reduction in area due to outburst, which is in agreement with evolutionary process observations (Figure 7a).

In general, the shrinkage of lake area in 1920−1995 is reflected in the spatial patterns. The comprehensive lake expansion in 1995−2010 is also in agreement with the evolution revealed by the time series. The overall

increase, albeit with variability in the 2010−2020 period, is visible in the spatial differences, with a clearer contrast through Landsat era (1970−2020).

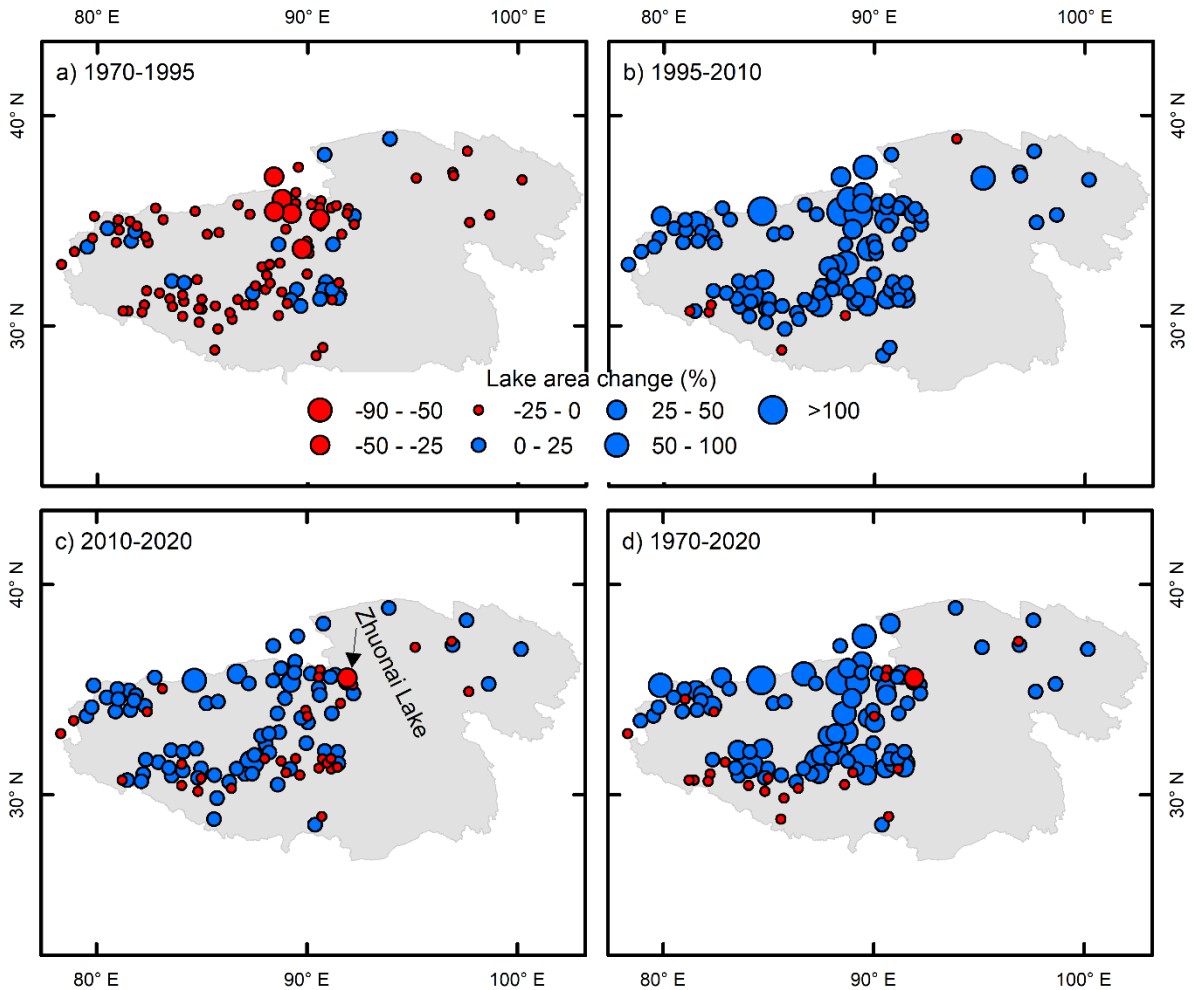

**Figure 10.** Spatial patterns of lake area changes during the Landsat era. a) 1970−1995. b) 1995−2010. c) 2010−2020. d) 1970−2020.


## 5   Discussion

### 5.1 Uncertainties in lake mapping

The uncertainties in lake mapping from topographic maps and Landsat images have usually been estimated by half a pixel inside and outside the delineated lake boundary (Fujita et al., 2009; Salerno et al.,

2012). However, for the 1920 lake mapping, such an uncertainty estimate is inappropriate because the process and principle of the map creation are not known. In addition, there are offsets of the maps for some lakes, as shown in Figure 7a, although a geometric correction has been conducted before delineation. In any case, such offsets should not have an impact on the calculation of lake area.

Having three diverse data sources with different spatial resolutions (Table 1) could lead to differences in lake area estimates. However, a long-term (100-year) lake evolution is examined, and lakes over the QTP have experienced a remarkable change. The inconsistency in spatial resolution between the data sets seems to have had little effect on altering the trend of lake evolution. The methods used for delineating lake boundaries are also different for the different data sources. Visual interpretation and manual digitization were used for the 1920 and 1960 maps, and for the 1970 Landsat MSS data. The results were cross checked to ensure a high-quality output of lake boundaries. For Landsat TM/ETM+/OLI, an automated water classification with the optimal threshold for NDWI images is used, with the purpose of providing highly efficient and consistent water extraction. We have examined the output from this process by comparing it with false color composites of the original Landsat images to ensure lake boundaries are correct despite cloud and snow contamination of the water body surface. Inevitably, some Landsat images outside the optimal season have had to be selected, although at a several-year interval for each epoch (Figure 4). However, few data from these seasons are used, and those images that are used contain few lakes. Similar conditions apply to the selection of ETM+ images when no suitable Landsat TM data are available. Again, the proportion of lakes covered by these ETM+ images is small compared to those covered by the Landsat TM data.

**5.2 Causes of lake changes**

Lake water balances for the Inner plateau as a whole (covering most of the lakes) (Zhang et al., 2017b) or for typical lake basins such as Selin Co (Zhou et al., 2015), Nam Co (Li et al., 2017), and Qinghai Lake (Zhao et al., 2017) have been quantitatively assessed. These studies reveal that increased precipitation, rather than glacier meltwater supply, has made the dominant contribution (>70%) to lake growth in recent decades. Further analysis of excess glacier meltwater runoff and lake volume increase in the Inner QTP (an endorheic basin with dominant lakes, Figure S1) shows that glacier melt contributes a small fraction to lake growth (~14%) between 2000 and 2019 (Shean et al., 2020; Zhang et al., 2021). This suggests a small role of glacier meltwater to lake status, although it is impossible to examine for last 100-year lake evolution.

Based on this assumption, a comparison of time series of precipitation with lake area can reveal the dominant driver of lake evolution. Comprehensive warming has occurred over the QTP during the past 100

years (Figure 11). The time series of temperature shows an increase followed by a decrease before ~1965 and then a warming trend until the present day. The spatial pattern of precipitation change indicates a wetting trend everywhere except for the southwestern and southeastern peripheries of the QTP. The time series of precipitation shows a drying trend between 1920 and 1995, followed by a wetting between 1995 and 2010 but

with variability after 2010 (Figure S3). The correlations (coefficient of determination) between temperature and lake area, cumulative precipitation and lake area are 0.58 and 0.80, respectively (Figure 11). The high relationship between cumulative precipitation and lake area implies the precipitation is the dominant contributor of lake area evolution. The spatial-temporal patterns of precipitation changes, especially the time-series, relate well with the evolution of total lake area (Figure 11, S3), suggesting that precipitation has mainly

controlled lake evolution for the past 100 years. Additionally, these data also give further weight to the argument that the lake development patterns before 1960 are reliable, although no field measurements or satellite observations are available to validate them.

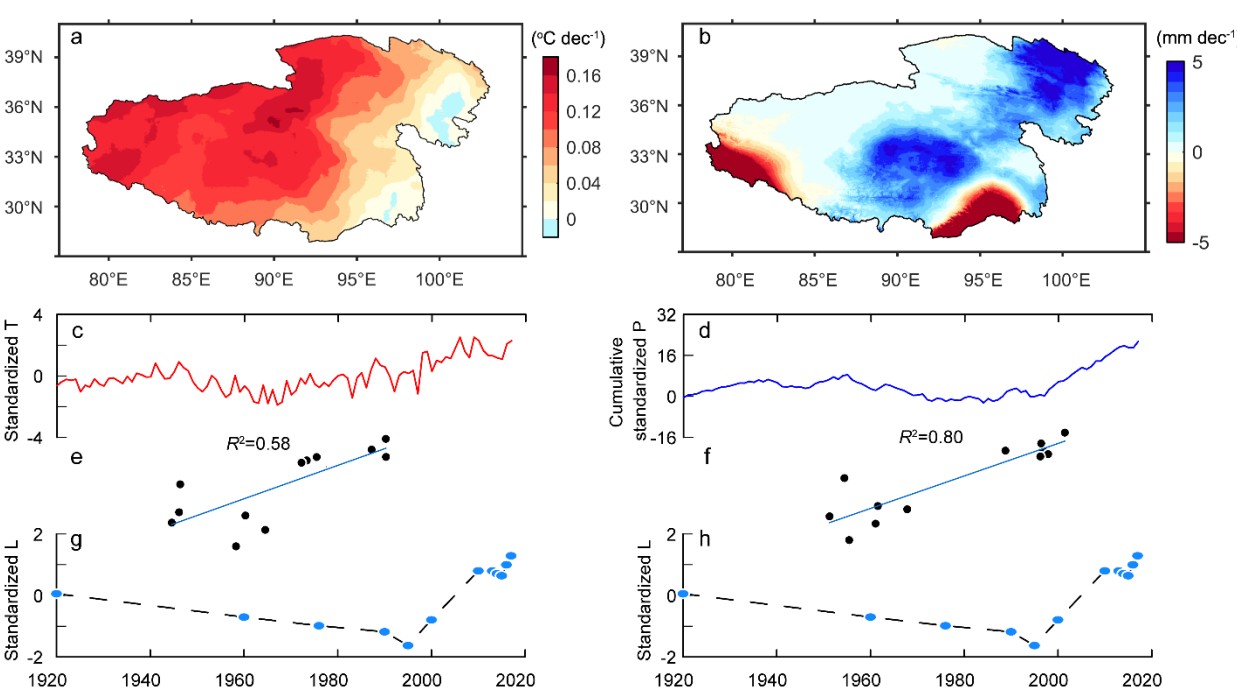

**Figure 11.** Spatial-temporal variations of temperature and precipitation from 1920 to 2017, and their correlations
with lake area changes. a) Spatial pattern of temperature change. b) Spatial pattern of precipitation change. c) Time series of standardized temperature (T). d) Time series of cumulative standardized precipitation (P). e) Correlation

between standardized temperature and standardized lake area. f) Correlation between cumulative standardized precipitation and standardized lake area. g−h) Standardized lake area.

**6 Data availability**

This lake data set includes the lake evolution during the past 100 years (from 1920 to 2020) over the QTP, which is defined by the boundaries of Tibet Autonomous Region and Qinghai province, with an area of about $200\times10^4$ km$^2$. The 1920 and 1960 lake mapping are derived from manual digitizing of an early map of Republic of China, and the topographic map of China, respectively. The continuous lake mappings from 1970

to 2020 are derived from Landsat MSS, TM, ETM+ and OLI images by a semi-automatic water classification. All lake boundaries have been visually checked against original maps/images. All the analysis presented in this study is based on this lake data set.

We also provide the lake data set for the period 1970 to 2020 for the TP, the boundary of which is defined by the altitude of 2500 m, and which has an area of about $300\times10^4$ km$^2$. The boundary of the TP was first

defined in this way by Zhang et al. (2013), and is popularly used. This lake data set, which includes more of the western plateau, is useful for more extensive analyses of climate and cryosphere variations.

Both lake data sets can be freely downloaded at http://doi.org/10.5281/zenodo.4678104 (Zhang et al., 2021). The lake data sets are provided as ArcGIS Shapefiles, which are easy to read or reanalyze in GIS environments.


**7 Conclusions**

In this study, we have compiled a lake data set for the QTP spanning the early 20th century to the early 21th century with a relatively high spatial resolution from map observations. This is the longest, and the most comprehensive, lake mapping available for the QTP. The lake products generated, including lakes in 1920 from

maps of the Republic of China, lakes in 1960 from the topographic map of China, and lakes for the period from 1970 through 2020 from 1477 Landsat images.

In 1920 there were 604 lakes larger than 1 km$^2$ over the QTP, with a total area of 40779.33 km$^2$. One hundred years later, the number has increased to 1197 (~98%) with a total area of 46201.62 km$^2$ (~13%). The

increase in numbers can probably be mainly attributed to the fact that lakes smaller than 1 km$^2$ have exceeded 1

km$^2$ over time and thus have been added to the count. Detailed lake evolution is also examined. The lakes

underwent a decrease in area over the period 1920−1995, followed by a rapid increase in 1995−1995. They

tended to stability in 2010−2015, and increased in area again in 2015−2020. The spatial patterns of overall

shrinkage or expansion are consistent with time series, but a pattern of enlargement for central-north lakes

against contraction for southern lakes is revealed, which is especially clear between 1970 and 2020. The time

series of precipitation is coincident with the trends of lake evolution.

This high-quality lake data set is a great asset for interdisciplinary climate, cryosphere, and hydrosphere

studies, and can even form a bridge linking together earth system science over the world's highest plateau. We

will update the lake data set in the future when further data become available, for example with annual updates

with new Landsat images.


**Supplement**

The supplement related to this article is available online at: XXX

**Author contributions:** GZ designed the study and wrote the manuscript. YR provided the lake data set for the

Republic of China in 1920, and WW provided the lake data set from the topographic map of China in 1960. All

authors contributed to the writing and editing of this paper.

**Competing interests:** The authors declare that they have no conflict of interest.

**Acknowledgements:** The first author would like to thank former students or team members, Kexiang Zhang,

Fujing Zhu, Guoxiong Zheng, for data/codes accumulation. We also thank Hsiung-Ming Liao at Academia

Sinica for providing maps of the Republic of China.

**Financial support:** This study was supported by Basic Science Center for Tibetan Plateau Earth System

(BSCTPES, NSFC project No. 41988101-03), grants from the Natural Science Foundation of China

(41871056, 41831177), the Second Tibetan Plateau Scientific Expedition and Research (STEP) program

(2019QZKK0201), and the Strategic Priority Research Program (A) of the Chinese Academy of Sciences

(XDA20060201).

**Review statement:** This paper was edited by Kirsten Elger and reviewed by two anonymous referees.

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
