# Peer review of "years of lake evolution over the Qinghai-Tibet Plateau"

_Earth System Science Data, 2021_

## Community Comment (CC1)

**Response**: We are grateful to the anonymous reviewer for the constructive comments on our manuscript. We have carefully addressed all the comments below.

This is a great dataset that expands 100 years for the lakes in Qinghai-Tibet Plateau. I appreciate the authors' effort to put all these data together in a consistent format and free available to the world. I knew many papers based on the Landsat images have been published, the 1960s data and a few other recent data have been also published, but it is the first time the 1920s data are publishing, also to combine three different types of data into one systematic dataset is awesome. The paper (data together) has revealed a general pattern of lake reduction from the 1920s to 1995, lake expansion from the 1995-2020. This further confirms my thought based on previous data and publications, but not yet published yet. I am very happy this paper did it. the overall writing of the paper is very good, although there are minor grammar/tense errors that can be fixed relatively easy. For example, the paper is primarily written in a present tense, but in some places mixed past and present tenses are used and should be fixed (see line 175 "was" should be "is").

**Response**: We thank the reviewer for the positive evaluation of our work. We have fixed the tenses of mixed use.

I do have one comment about the 1920 maps/data. I fully understand that it is impossible to validate the quality and accuracy of the data now. But I hope if any possible to give some details on how they actually mapped such a big area, with little or no current technology,..., in some places, these types of information should be there. I know it is hard to find but just curious. Also although the number of lakes (larger than 1 km2) was only 710, but the total area of these lakes are larger than the 1960s (1107 lakes). My guess would be many small lakes not in their map might be just missed due to the reason they really did not see them during the field work, since it was really a big task to walk through all small lakes (just my thoughts). However, the missing small lakes do not impact their results, since the major and big lakes really contribute to the total lake area, which was indeed larger than 1960s. one curious question is that, have authors checked all the same 710 lakes in the 1920, 1960, 1970, ...., 2020? Can you just show the trend and pattern for these 710 lakes? How many lakes in Figure 8?

**Response**:
We have added some sentences below in Section "3 Data and methods" for the produce of Maps of the Republic of China in 1920.
"*Maps of the Republic of China used in this study are from "1/500 000 map of China", which is mainly surveyed and drawn by Beijing Army Survey Bureau, Staff Headquarters Cartographic Bureau, and Guangdong Army Survey Bureau in 1916–1918 (http://www.ccartoa.org.tw/news/2018/180901.html) (Table 1). The Army Survey Bureau of the Republic of China made the "Ten-Year Rapid Survey Plan", which required all provinces to complete topographic maps of different scales as soon as possible. The Maps of the Republic of China were finished at beginning to apply surveying and mapping techniques, which was the first national basic surveying and mapping program after the establishment of the General Administration of Army Surveying of the General Staff Headquarters of the Republic of China in 1914. The maps were compiled with 1º in latitude by 1º in longitude square grid, a regional independent (hypothetical) coordinate system in each province such as polyhedral projection (modified multiconic projection), and multicolor printing. The actual measurement of astronomy, triangulation, and barometric altimetry were utilized to provide a plane and elevation measurement control basis for topographic survey. This is the most accurate topographic map with the most complete coverage available from the early 20th century...*"

We did not show the same lakes in the 1920, 1960, 1970, ...., 2020 as the offset for some lakes (especially for some small lakes) in early 1920 and 1960 maps as well as some new formed and disappeared lakes. **However, we added a new figure (Figure 6) to show the mapped lakes in 1920, 1960, and 2020.** Therefore, we showed the total area evolution for lakes greater than 1 km$^2$ in the entire the Qinghai-Tibet Plateau, and number for large lakes (>100 km$^2$ each lake) with relative stable evolution (small lakes with a large fluctuation in number). We only showed 66 large lakes (greater than 50 km$^2$ each lake among 1920, 1995 and 2020) in Figure 8.

[Figure]

*Figure 6 (new). Mapped lakes in 1920 from maps of the Republic of China, 1960 from Topographic map of China, and 2020 from Landsat 8 images.*

Figure 6, I see some red outlines, some clearly are the lake boundaries, but others are not. Please make sure they are cleaned...

**Response**: We have removed the read lines in Figure 6a.

Line 305, add "due to outburst" behind "a noticeable reduction in area"

**Response**: Corrected

[revised manuscript text omitted]

---

## Author Response (AR1)

**Response to Referee #1**

**Response**: We are grateful to the anonymous reviewer for the constructive comments on our manuscript. We have carefully addressed all the comments below.

This is a great dataset that expands 100 years for the lakes in Qinghai-Tibet Plateau. I appreciate the authors' effort to put all these data together in a consistent format and free available to the world. I knew many papers based on the Landsat images have been published, the 1960s data and a few other recent data have been also published, but it is the first time the 1920s data are publishing, also to combine three different types of data into one systematic dataset is awesome. The paper (data together) has revealed a general pattern of lake reduction from the 1920s to 1995, lake expansion from the 1995-2020. This further confirms my thought based on previous data and publications, but not yet published yet. I am very happy this paper did it. the overall writing of the paper is very good, although there are minor grammar/tense errors that can be fixed relatively easy. For example, the paper is primarily written in a present tense, but in some places mixed past and present tenses are used and should be fixed (see line 175 "was" should be "is").

**Response**: We thank the reviewer for the positive evaluation of our work. We have fixed the tenses of mixed use.

I do have one comment about the 1920 maps/data. I fully understand that it is impossible to validate the quality and accuracy of the data now. But I hope if any possible to give some details on how they actually mapped such a big area, with little or no current technology,…, in some places, these types of information should be there. I know it is hard to find but just curious. Also although the number of lakes (larger than 1 km2) was only 710, but the total area of these lakes are larger than the 1960s (1107 lakes). My guess would be many small lakes not in their map might be just missed due to the reason they really did not see them during the field work, since it was really a big task to walk through all small lakes (just my thoughts). However, the missing small lakes do not impact their results, since the major and big lakes really contribute to the total lake area, which was indeed larger than 1960s. one curious question is that, have authors checked all the same 710 lakes in the 1920, 1960, 1970, …., 2020? Can you just show the trend and pattern for these 710 lakes? How many lakes in Figure 8?

**Response**:
We have added some sentences below in Section "3 Data and methods" for the produce of Maps of the Republic of China in 1920.
"*Maps of the Republic of China used in this study are from "1/500 000 map of China", which is mainly surveyed and drawn by Beijing Army Survey Bureau, Staff Headquarters Cartographic Bureau, and Guangdong Army Survey Bureau in 1916–1918 (http://www.ccartoa.org.tw/news/2018/180901.html) (Table 1). The Army Survey Bureau of the Republic of China made the "Ten-Year Rapid Survey Plan", which required all provinces to complete topographic maps of different scales as soon as possible. The Maps of the Republic of China were finished at beginning to apply surveying and mapping techniques, which was the first national basic surveying and mapping program after the establishment of the General Administration of Army Surveying of the General Staff Headquarters of the Republic of China in 1914. The maps were compiled with 1° in latitude by 1° in longitude square grid, a regional independent (hypothetical) coordinate system in each province such as polyhedral projection (modified multiconic projection), and multicolor printing. The actual measurement of astronomy, triangulation, and barometric altimetry were utilized to provide a plane and elevation measurement control basis for topographic survey. This is the most accurate topographic map with the most complete coverage available from the early 20th century…*"

We did not show the same lakes in the 1920, 1960, 1970, …., 2020 as the offset for some lakes (especially for some small lakes) in early 1920 and 1960 maps as well as some new formed and disappeared lakes. **However, we added a new figure (Figure 6) to show the mapped lakes in 1920, 1960, and 2020.** Therefore, we showed the total area evolution for lakes greater than 1 km$^2$ in the entire the Qinghai-Tibet Plateau, and number for large lakes (>100 km$^2$ each lake) with relative stable evolution (small lakes with a large fluctuation in number). We only showed 66 large lakes (greater than 50 km$^2$ each lake among 1920, 1995 and 2020) in Figure 8.

[Figure]

*Figure 6 (new).* *Mapped lakes in 1920 from maps of the Republic of China, 1960 from Topographic map of China, and 2020 from Landsat 8 images.*

Figure 6, I see some red outlines, some clearly are the lake boundaries, but others are not. Please make sure they are cleaned…

**Response**: We have removed the read lines in Figure 6a.

Line 305, add "due to outburst" behind "a noticeable reduction in area"

**Response**: Corrected

**Response to Referee #2**

The paper by Chang et al. deals with change detection of lake coverage on the Qinhai-Tibetan Plateau during the last hundred years. As one commentator already pointed out in the public discussion, this is an amazing paper. I am not so much familiar with GIS approaches, but my impression is that the authors did a sophisticated job and considered all methodic limitations. The results are very impressive and might also be of high value for palaeoclimatologists dealing with ancient lake dynamics. I agree with the public review. In addition, the following issues should be considered in a revised manuscript:

**Response**: We thank the reviewer for the positive evaluation of our work. We have carefully addressed all the comments below.

Line 41: Degradation of „thawing" permafrost; snow and ice melts.

**Response**: Many thanks for the suggestion, and we have changed this sentence as:

*"Cryospheric melting reflects in accelerated glacier retreat and ice loss (Yao et al., 2012; Shean et al., 2020; Hugonnet et al., 2021), a lower snowline (Shu et al., 2021), thawing permafrost (Ran et al., 2018), and snow melt (Pulliainen et al., 2020),…"*

Lines 60-62: „….no studies of lake mapping…" versus „…have been considered in several studies…" Sounds like a contradiction, needs rephrasing.

**Response**: "… no studies of lake mapping…" This corresponds to the Republic of China (the early 20th centuries), however, "…have been considered in several studies…" is the present state (mainly satellite era, 1960s-). We changed two sentences as:

*"However, no studies have yet reported lake mapping for the remote QTP in the Republic of China (the early 20th centuries).*
*Changes in present lake number and area (mainly satellite era, 1960s−) have been considered in several studies: …"*

Lines 82-85: The political assignment of geographic terms by countries and the international community sometimes differ. The region not „sometimes", but „often" is referred to as Tibetan Plateau. Just use your terminology and definition of the area. This is fine, but rephrasing is needed to understand what you mean.

**Response**: We agree with the reviewer, this could make reader confuse. We simplified these sentences greatly as:

*"The QTP, with an area of ~200×10$^4$ km$^2$, consists of Tibet Autonomous Region and Qinghai province, and has the border of China as its southern boundary (Figure 1). The boundary of QTP is different from the usually named as the Tibetan Plateau (TP, the area with an altitude above 2500 m a.s.l.) (Figure S1)…."*

I generally appreciate the high quality and layout of the illustrations. In Fig. 6, the old shore lines in red and yellow are difficult to note and should be mentioned in the caption.

**Response**: The Reviewer #1 also mentioned this problem. We removed red lines in Figure 6a (now Figure 7a).

Causes of lake change are well discussed. Maybe, it is possible to give some values of correlation between climate parameters and average lake parameters (lake numbers, lake area through time). I would also be good to have values for glacial melt-water runoff to substantiate its unimportant role for lake status. Could be added to Figure 10. The latter figure should also include an average curve of lake development.

**Response**: We thank the reviewer for these suggestions. We added the correlations between lake area and temperature ($R^2$=0.58), and between lake area and cumulative precipitation ($R^2$=0.80) in Figure 10 (now Figure 11, below). The lakes over the QTP are mainly located in the endorheic Inner Plateau without water outflowing (Figure S1). The geodetic method estimated the excess glacier meltwater runoff in the Inner Plateau between 2000 and 2018 is 1.12±0.29 Gt/yr (Shean et al., 2020). In addition, the lake volume increase rate in the Inner Plateau between 2000 and 2019 is 7.79 Gt/yr. Therefore, we can estimate that glacier meltwater runoff contribution to lake storage gain between 2000 and 2019 is approximately 14%. We added several sentences below in Discussion section.

"*Further analysis of excess glacier meltwater runoff and lake volume increase in the Inner QTP (an endorheic basin with dominant lakes, Figure S1) shows that glacier melt contributes a small fraction to lake growth (~14%) between 2000 and 2019 (Zhang et al., 2021; Shean et al., 2020). This suggests a small role of glacier meltwater to lake status, although it is impossible to examine for last 100-year lake evolution.*"

[Figure]

**Figure 11.** Spatial-temporal variations of temperature and precipitation from 1920 to 2017, and their correlations with lake area changes. a) Spatial pattern of temperature change. b) Spatial pattern of precipitation change. c) Time series of standardized temperature (T). d) Time series of cumulative standardized precipitation (P). e) Correlation between standardized temperature and standardized lake area. f) Correlation between cumulative standardized precipitation and standardized lake area. g–h) Standardized lake area.

**References**

Shean, D. E., Bhushan, S., Montesano, P., Rounce, D. R., Arendt, A., and Osmanoglu, B.: A Systematic, Regional Assessment of High Mountain Asia Glacier Mass Balance, Frontiers in Earth Science, 7, 363, 10.3389/feart.2019.00363, 2020.

Zhang, G., Bolch, T., Chen, W., and Crétaux, J. F.: Comprehensive estimation of lake volume changes on the Tibetan Plateau during 1976–2019 and basin-wide glacier contribution, Science of the Total Environment 772, 145463, 10.1016/j.scitotenv.2021.145463, 2021.